# Toll-like Receptor-9 (TLR-9) Signaling Is Crucial for Inducing Protective Immunity following Immunization with Genetically Modified Live Attenuated *Leishmania* Parasites

**DOI:** 10.3390/pathogens12040534

**Published:** 2023-03-29

**Authors:** Parna Bhattacharya, Sreenivas Gannavaram, Nevien Ismail, Ankit Saxena, Pradeep K. Dagur, Adovi Akue, Mark KuKuruga, Hira L. Nakhasi

**Affiliations:** 1Division of Emerging and Transfusion Transmitted Disease, Center for Biologics Evaluation and Research Food and Drug Administration, Silver Spring, MD 20993, USA; 2Immune Monitoring Shared Resource, Rutgers, Cancer Institute of New Jersey, New Brunswick, NJ 08901, USA; 3Flow Cytometry Core, National Heart, Lung, and Blood Institute, National Institutes of Health, Bethesda, MD 20892, USA; 4Division of Bacterial, Parasitic, and Allergenic Products, Center for Biologics Evaluation and Research Food and Drug Administration, Silver Spring, MD 20993, USA

**Keywords:** visceral leishmaniasis, dendritic cells, myeloid differentiation primary response 88, nuclear factor-kappa B, cytokines, costimulatory molecules, innate immunity

## Abstract

No human vaccine is available for visceral leishmaniasis (VL). Live attenuated centrin gene-deleted *L. donovani* (*LdCen^−/−^*) parasite vaccine has been shown to induce robust innate immunity and provide protection in animal models. Toll-like receptors (TLRs) are expressed in innate immune cells and are essential for the early stages of *Leishmania* infection. Among TLRs, TLR-9 signaling has been reported to induce host protection during *Leishmania* infection. Importantly, TLR-9 ligands have been used as immune enhancers for non-live vaccination strategies against leishmaniasis. However, the function of TLR-9 in the generation of a protective immune response in live attenuated *Leishmania* vaccines remains unknown. In this study, we investigated the function of TLR-9 during *LdCen^−/−^* infection and found that it increased the expression of TLR-9 on DCs and macrophages from ear-draining lymph nodes and spleen. The increase in TLR-9 expression resulted in changes in downstream signaling in DCs mediated through signaling protein myeloid differentiation primary response 88 (MyD88), resulting in activation and nuclear translocation of nuclear factor-κB (NF-κB). This process resulted in an increase in the DC’s proinflammatory response, activation, and DC-mediated CD4^+^T cell proliferation. Further, *LdCen^−/−^* immunization in TLR-9^−/−^ mice resulted in a significant loss of protective immunity. Thus, *LdCen^−/−^* vaccine naturally activates the TLR-9 signaling pathway to elicit protective immunity against virulent *L. donovani* challenge.

## 1. Introduction

Visceral leishmaniasis (VL) is caused by the protozoan parasite *Leishmania donovani*, which affects nations worldwide. An estimated 50,000–90,000 new VL cases are recorded every year from endemic areas of the world including Bangladesh, India, Nepal, Brazil, Ethiopia, Kenya, and Sudan [1].

Host defenses against *Leishmania* species are mainly initiated by innate immune cell-induced activation of an inflammatory response, followed by cell-mediated immune responses [2]. The innate immune system actively contributes to the rapid identification of pathogens, such as viruses and bacteria, by using a range of pattern recognition receptors (PRRs), such as Toll-like receptors [3,4,5,6,7]. TLRs are cellular receptors that detect common pathogen-associated molecules such as lipopeptides, peptidoglycan, flagellin, lipopolysaccharides, and nucleic acids and induce innate responses by several pathways, facilitating the generation of inflammatory cytokines by macrophages and dendritic cells (DCs) [8,9]. During *Leishmania* infection, the participation of TLR-9, TLR-4, TLR-2, and TLR-3 is pivotal for mediating a proinflammatory cytokine response and subsequent infection control [10,11,12,13].

The findings that resistance to *Leishmania* is dependent on parasite lipophosphoglycan (LPG) by TLR-2 [14], the generation of IL-12, and the establishment of a Th1 immune response [15] in addition to NO production have validated the significance of TLR-2 in infection control [16]. Previous studies also have demonstrated a significant function for TLR-2/TLR-3 in *L. donovani* recognition and parasite clearance by macrophages via secretion of nitric oxide (NO) and proinflammatory cytokines [13]. Similarly, TLR-4 signaling controls *L. major* parasite growth, and lack of TLR-4 signaling exacerbates parasite growth during both the innate and adaptive phase of the immune response as well as a delay in the healing of the cutaneous wounds [17,18]. Recent studies also provided novel insights into the mechanism of TLR-9-induced host protection during *Leishmania* infection. For example, TLR-9 recognizes CpG DNA sequences of *L. major* and activates the dendritic cell (DC) along with the generation of a Th1-dominant response that resolves the lesions [19]. Likewise, during *L. infantum* infection, TLR-9 activates CD11c^high^ DCs thereby triggering IFN-γ production and cytotoxic function in NK cell [20,21]. Notably, enhancing the effectiveness of immunization with parasite antigens against leishmaniasis by activating TLRs via exogenous agonists has emerged as a viable option [10,22,23].

Of the numerous vaccination methods against *Leishmania*, live attenuated vaccines showed great potential because of mimicking the natural infection [24,25,26]. Our laboratory created several *Leishmania* mutant strains including *L. donovani (LdCen^−/−^),* which lack the Centrin1 gene [27,28,29,30]. We showed that immunization with *LdCen^−/−^* resulted in protection against virulent *L. donovani* infection in a variety of animal models, including mice, hamsters, and dogs. Control of parasite burden and induction of an adaptive T cell response that is host-protective served as indicators of *LdCen^−/−^* induced protection [31,32,33,34]. We recently reported that immunization with *LdCen^−/−^* parasites and innate cells such as neutrophils, macrophages, and DCs are critical for eliciting a protective Th1 immune response [28,35,36,37]. Innate cell-mediated recognition of pathogens and subsequent initiation of signal transduction pathways that further instruct the development of antigen-specific adaptive immunity mostly depend on TLRs [38]. Several TLR ligands have been utilized as promising immune enhancers for vaccination against VL [39]. Specifically, TLR-9 ligand, CpG-ODN, has been extensively explored as a preventative vaccine adjuvant for *Leishmania* antigens and it has been demonstrated to provide protection following challenges with *L. major* and *L. donovani* [39,40,41,42,43,44].

Considering the beneficial effects of TLR-9 activation in *Leishmania* vaccine studies, we have investigated the role of TLR-9 in *LdCen^−/−^* induced immunity. We have demonstrated that *LdCen^−/−^* infection specifically upregulates TLR-9 expression and subsequent TLR-9-mediated downstream signaling in DCs compared to *LdWT* infection. These events enable DCs to produce an increased proinflammatory response and initiate CD4^+^Th1 cell proliferation. Importantly, significant loss of protection by *LdCen^−/−^* immunization in TLR-9^−/−^ mice against virulent *L. donovani* challenge highlights the crucial role of TLR-9 activation for the generation of a host-protecting immune response by *LdCen^−/−^* vaccine. 

## 2. Material and Methods

### 2.1. Animals and Parasites

Female C57BL/6 mice aged five to six weeks were purchased from The National Cancer Institute, National Institutes of Health, Bethesda, MD; TLR-9^−/−^ mice were purchased from Jackson Laboratory. We performed all our tests on female mice that were 6 to 8 weeks old. Mice were kept in an environment that was appropriate for this species at the Food and Drug Administration/CBERAAALAC-accredited facility. The *LdWT* (MHOM/SD/62/1S) and *LdCen^−/−^* line of *L. donovani* (Ld1S2D) parasites were employed, and the parasite culture process and standard molecular biology practices were carried out as previously reported [31].

### 2.2. Ethics Statement

The study’s animal protocol (ASP 1995#26) received clearance from the Institutional Animal Care and Use Committee of the Center for Biologics Evaluation and Research of the Food and Drug Administration. Further, the animal protocol is in full accordance with the “Guide for the Care and Use 198 of Laboratory Animals” as described in the U.S. Public Health Service 199 Policy on Humane Care and Use of Laboratory Animals 2015. (http://grants.nih.gov/grants/olaw/references/phspolicylabanimals.pdf; accessed on 15 October 2021).

### 2.3. Infection of Mice and Isolation of Macrophages and DCs from Ear dLN and Spleen

Using a 29-gauge needle (BD Ultra-Fine) and a volume of 10 μL, 3 × 10^6^ *LdWT/LdCen*^−/−^ parasites were intradermally (ID) injected into C57BL/6 mice in the ear pinna. Each study employed a minimum of six mice per group. As a control group, naïve mice that were age-matched received PBS. Macrophages (MØ) (Cd11b^+^Ly6G^−^Ly6c^−^Cd11c^−^MHCII^+^) and dendritic cells (DC) (Cd11b^+^Ly6G^−^Ly6c^−^Cd11c^+^MHCII^hi^) were sort selected from the spleen and ear dLN of *LdWT* and *LdCen^−/−^* infected mice at 72 h and 7 days after infection, respectively, by high-speed FACS cell sorter system (BD FACS ARIA IITM). Briefly, tweezers and a syringe plunger were used to physically separate and remove the retro maxillary (ear-draining) lymph nodes. Filtration of tissue homogenates using a 70-μm cell strainer was performed (Falcon Products, Corning, NY, USA). To create a single-cell suspension, mouse spleens were collected and processed with collagenase (1 mg/mL) and DNase I (20 mg/mL) (Thermo Fisher Scientific, Waltham, MA, USA). The single cell suspension from the spleen and dLN of the ear was labeled with anti-TCR-, anti-NK1.1, and anti-Cd1b that had been APC-tagged. To exclude certain cell types, anti-APC magnetic beads were utilized and run across LS columns. Flow through enriched population of macrophages and DCs were collected and stained with macrophages and DC-specific markers and then further sorted.

### 2.4. Cultivation of Bone Marrow-Derived Dendritic Cells (BMDCs)

In vitro dendritic cell culture was performed using bone marrow progenitors. Briefly, mice’s tibias and femurs were removed, cleared of tissue, and then cleansed with RPMI media. The erythrocytes were removed using ACK lysis buffer (Lonza, Rockville, MD, USA), and the isolated bone marrow was then cultured for 7 days with complete RPMI medium supplemented with 10% (*v*/*v*) fetal bovine serum (FBS) (R&D systems, Minneapolis, MN, USA), 1% penicillin (20 U/mL)/streptomycin (20 g/mL) (Thermo Fisher Scientific, Waltham, MA, USA ), and 20 ng/mL GM-CSF (Peprotech, Cranbury, NJ, USA) and IL-4 (Peprotech, Cranbury, NJ, USA) for obtaining >75% purity of DCs and analyzed by flow cytometry. BMDCs were transfected with TLR-9 siRNA/MyD88 siRNA or treated with/without JSH-23 (NF-κB transcriptional activity inhibitor) (Abcam, Waltham, MA, USA) followed by infection. In some experiments BMDCs were treated with LPS (Sigma, St. Louis, MO, USA) (1 μg/mL) for 45 min before siRNA transfection/JSH-23 treatment and infection. By using sandwich ELISA, mouse cytokines were detected in the conditioned medium of BMDC cultures. Using a sandwich ELISA kit (Thermo Fisher Scientific, Waltham, MA, USA), culture supernatants were collected 24 h after infection to assess cytokine production. Using flow cytometry, acquisitions of a million events were carried out to quantify the expression of costimulatory molecules on the surface of DCs. The flow cytometry section of the Material and Methods has a full description of the technique and antibodies.

### 2.5. TLR-9 siRNA and MyD88 siRNA Mediated Silencing in DC In Vitro

TLR-9 siRNA and control siRNA were purchased from Thermo Fisher Scientific (Waltham, MA, USA). MyD88 siRNA and control siRNA were purchased from Santa Cruz Biotechnology Inc. (Santa Cruz, CA, USA). Cell transfections with siRNAs were performed according to the manufacturer’s protocol.

### 2.6. NF-κB Nuclear Translocation

DCs were isolated from mouse BM and stimulated with LPS (1 μg/mL) for 45 min followed by infection with *LdWT/LdCen^−/−^* parasites and recovered for analysis. After 6 h of incubation, Fc receptors were blocked with normal mouse serum for 15 min at 4 °C, and cells were fixed by 4% formaldehyde for 10 min at room temperature. Then, DCs were treated with monoclonal rabbit anti-mouse pNF-κBp65 (Cell signaling, Danvers, MA, USA) at a dilution of 1:50 in permeabilization buffer for 30 min at room temperature. Cells were then stained using a 1:200 dilution of an Alexa 647-conjugated goat anti-rabbit secondary antibody (Thermo Fisher Scientific, Waltham, MA, USA) for 30 min at room temperature. The cells were stained with DAPI (Thermo Fisher Scientific, Waltham, MA, USA) for the nucleus before being captured using Image stream AMNIS. By using the TransAM NF-κB activation assay kit (Active Motif, Carlsbad, CA, USA) NF-κBp65 activation in LPS-treated control and infected DCs was also quantitatively evaluated as per manufacturer’s instructions.

### 2.7. Antigen Presentation Assay: In Vitro DC and T Cell Co-Culture Studies

After being transfected with control siRNA/TLR-9 siRNA or treated or untreated with JSH-23 and an OVA peptide pulse (2 μg/mL; residues 323 to 339; Anaspec, Fremont, CA, USA), DCs were infected for 24 h with either *LdWT* or *LdCen^−/−^* parasites. From the spleens of DO11.10 transgenic mice, CD4^+^ T lymphocytes were isolated and stained with 5 μM carboxyfluorescein succinimidyl ester (CFSE) (Thermo Fisher Scientific, Waltham, MA, USA) for 10 min in RPMI 1640 without fetal calf serum (FCS) at 37 °C in a 5% CO_2_ humidified chamber. After that, cells were incubated with ice-cold RPMI 1640 and 10% FCS for 5 min to quench the CFSE, and cells were properly washed before being plated in 96-well tissue culture plates with OVA-pulsed BMDC. CD4 T cell proliferation was then calculated using flow cytometry by gating on CD4^+^T cells after 5 days at 37 °C with 5% CO_2_. An amount of 10,000 CD4-positive cells were counted in each sample. The program utilized was FlowJo version 9.7.5. Day 5 culture supernatants were collected for an ELISA test using a sandwich ELISA kit to assess cytokines (Thermo Fisher Scientific, Waltham, MA, USA). The assay was carried out in accordance with the manufacturer’s thorough instructions.

### 2.8. RT-PCR

Total RNA was extracted from the (1) macrophages and DCs recruited in ear dLN or spleen following ID injection of either PBS/*LdWT* or *LdCen^−/−^* parasites (2) DCs (3) spleen utilizing RNAqueous-Micro kit (AM1931; Ambion, Austin, Texas, USA), which additionally removes any contaminating DNA by using on-column PureLink DNase treatment during RNA purification. A high-capacity cDNA reverse transcription kit from Applied Biosystems was used to reverse transcribe aliquots (400 ng) of total RNA into cDNA via random hexamers. The TaqMan gene expression master mix and prepared TaqMan gene expression assays (Applied Biosystems) were used with a CFX96 Touch Real-Time System (Bio- Rad, Hercules, CA, USA) to measure the levels of cytokine gene expression. CFX Manager Software was used to evaluate the data. The CFX96 Touch Real-Time System was used to measure the expression of the following genes: TLR-2 (Mm00442346_m1); TLR-4 (Mm00445273_m1); TLR-9 (Mm00446193_m1); MyD88 (Mm00440338_m1); and GAPDH (Mm99999915_g1). The 2-DD Cycle threshold approach was used to determine the expression values. Samples were compared to the expression levels from untreated samples or animals that had received PBS injections, as necessary, after being normalized to GAPDH expression.

### 2.9. Flow Cytometry

During the surface labeling of BMDCs, rat anti-mouse CD16/32 (BD Biosciences) from BD Pharmingen was used to block cells for 20 min at 4 °C (5 μg/mL). Cells were then stained with the following antibodies: anti-mouse CD11b (Thermo Fisher Scientific, Waltham, MA, USA), anti-mouse CD3 (Thermo Fisher Scientific, Waltham, MA, USA), anti-mouse CD4(Biolegend, San Diego, CA, USA), anti-mouse Cd44 (Thermo Fisher Scientific, Waltham, MA, USA), anti-mouse Cd11c (Thermo Fisher Scientific, Waltham, MA, USA), anti-mouse Ly6G (Thermo Fisher Scientific, Waltham, MA, USA), anti-mouse Ly6C (Biolegend, San Diego, CA, USA), anti-mouse CD80 (BD Bioscience, San Jose, CA, USA), anti-mouse MHCII Class II (I-A/I-E) (Thermo Fisher Scientific, Waltham, MA, USA), anti-mouse CD40 (Thermo Fisher Scientific, Waltham, MA, USA), and anti-mouse CD200 (Thermo Fisher Scientific, Waltham, MA, USA); each with 1:100 dilution at 4 °C. All flow studies’ samples were stained with Live/Dead Fixable Aqua (Thermo Fisher Scientific, Waltham, MA, USA) to mark the dead cells. Cells were then fixed using a Fixation/Permeabilization Solution Kit (BD Bioscience, San Jose, CA, USA) for 20 min at room temperature after being washed twice with wash buffer. Cells were then acquired utilizing FACS Diva 6.1.2 software on an LSR II (BD Biosciences, San Jose, CA, USA) outfitted with laser lines of 407, 488, 532, and 633 nm. There were one million events acquired. Version 9.7.5 of the FlowJo program was used to analyze the data (Tree Star). First doublets were eliminated using the width parameter, and dead cells were disregarded based on Live/Dead Aqua dye (Thermo Fisher Scientific, Waltham, MA, USA) staining. Lymphocytes were classified using the characteristics of their light scattering. CD4 T cells were identified as CD3^+^ lymphocytes that express CD4 exclusively.

### 2.10. Immunization and Challenge Studies

Wild type (WT) (n = 13) or TLR-9^−/−^ (n = 13) mice were immunized intradermally with 3 × 10^6^ stationary-phase *LdCen^−/−^* promastigotes. On day 22, three animals from each experimental group were euthanized and the spleens were collected, and TLR-9 mRNA expression in splenocytes was evaluated by RT-PCR. On day 22, 10^5^ virulent *L. donovani (LdWT)* metacyclic parasites were administered by tail vein to the remaining animals (n = 6). Density gradient centrifugation was used to separate *L. donovani’s* infectious stage metacyclic promastigotes from stationary cultures as described before [45]. Age-matched naïve mice used as controls received 10^5^ virulent *L. donovani* metacyclic parasites in a similar manner. In order to quantify the parasite burden in the challenged mice’s spleens and livers at 6 weeks after the challenge, the separated host cell preparations were cultured using limiting dilutions as previously reported [31]. By using sandwich ELISA, IFN-γ and IL-10 were also detected in the *Leishmania* Ag-stimulated splenocyte culture supernatants 3 weeks after vaccination and 6 weeks after the challenge (Thermo Fisher Scientific, Waltham, MA, USA).

### 2.11. Statistical Analysis

GraphPad Prism 5.0 software was used to conduct an unpaired, two-tailed Student *t*-test to statistically analyze the differences in group mean values. A *p* value of 0.05 was regarded as statistically significant, and a *p* value of 0.005 was regarded as highly significant.

## 3. Results

### 3.1. Infection with LdCen^−/−^ Induces TLR-9 mRNA Expression

TLR-2, 4, and 9 are primarily responsible for influencing the various immune reactions during *Leishmania* infections [10,46]. Hence, we investigated the expression of TLR- 2, 4, and 9 in ear dLN and spleen-derived macrophages and dendritic cells from mice infected with *LdCen^−/−^* and compared it to *LdWT-*infected mice by real-time-PCR. Macrophages (MØ) (Cd11b^+^Ly6G^−^Ly6c^−^Cd11c^−^MHCII^+^) and dendritic cells (DC) (Cd11b^+^Ly6G^−^Ly6c^−^Cd11c^+^MHCII^hi^) were sort selected from ear dLN and the spleen of mice infected with *LdWT* and *LdCen^−/−^* at 72 h and 7 d post infection, respectively, and the expression profiles of TLR- 2, 4, and 9 were assessed. The representative sorting strategy has been displayed in Figure 1A,B showing the percentage of the positive population of macrophages and DCs from ear dLN of *LdWT* and *LdCen^−/−^* infected mice. TLR-9 expression was significantly higher in macrophages (Figure 1C,E) and DCs (Figure 1D,F) isolated from *LdCen^−/−^* infected mice’s ear dLN/spleen than in *LdWT-*infected animals. No significant difference in the expression of TLR-4 in these infections was observed. TLR-2 expression was significantly lower in macrophages and DCs from *LdCen^−/−^* infected mice compared to *LdWT-*infected mice (Figure 1C–F). These results demonstrate that there is an early induction of TLR-9 in the phagocytic cells following *LdCen^−/−^* infection.

### 3.2. LdCen^−/−^ Infected DCs Showed Heighted NF-κB Activation and Proinflammatory Cytokine Response via TLR-9- Myd88 Pathway

Since infection with *LdCen^−/−^* enhanced TLR-9 expression compared to *LdWT* infection, we investigated whether the TLR-9 mediated downstream signaling is also altered in *LdCen*^−/−^ infected DCs. We have specifically chosen DCs for our experiment since it has been demonstrated to be crucial in coordinating the immune response in *Leishmania* infection [47]. Downstream TLR-9 signaling involves Myd88-dependent pathway-mediated regulation of NF-κB transcription factor activation, which controls the expression of proinflammatory mediators [10]. Hence, we first assessed the translocation of NF-κB in *LdCen^−/−^* infected DCs either untreated or transfected with TLR-9 siRNA/Myd88-siRNAs or treated with JSH-23, an inhibitor of NF-κB transcriptional activity. Upon transfection with TLR-9 siRNA/Myd88-siRNAs, expression levels of TLR-9 and Myd88 mRNA were significantly reduced in *LdWT/LdCen^−/−^* infected DCs compared to untreated DCs, and the representative data for the *LdCen^−/−^* infected group has been shown (Figure 2A). Quantitative ELISA indicated marked NF-κBp65 activation in *LdCen^−/−^* infected DCs with a 1.2-fold increase at 6 h post infection compared to *LdWT* (Figure 2B). Further, Image Flow analysis of DCs showed an active translocation of immunofluorescence of p65 from the cytoplasm to the nucleus (Figure 2C). Silencing of either TLR-9 or MyD88 using corresponding siRNAs, respectively, or inhibition of NF-κB signaling using JSH-23, significantly abrogated NF-κB p65 activation and translocation in *LdCen^−/−^* infected DCs (Figure 2B,C). Active translocation of NF-κB to the nucleus facilitates up-regulation of the proinflammatory response [48]. Corroborating the heightened NF-κB activation and nuclear translocation in *LdCen^−/−^* infected DCs, we observed considerably increased production of proinflammatory cytokines IL-12 (Figure 2D) and TNF-α (Figure 2E) than in *LdWT* infected DCs. Importantly, silencing of TLR-9 or Myd88 or treatment with JSH-23 significantly abrogated the expression of IL-12 and TNF-α in *LdCen^−/−^* infected DCs (Figure 2D,E). These data show that *LdCen^−/−^* infection elicits the TLR-9-Myd88 signaling pathway to regulate inflammatory cytokine responses through the activation of the NF-κB.

### 3.3. TLR-9 Silencing or Inhibition of NF-κB Transcriptional Activity Significantly Reduced the Expression of Costimulatory Molecules While Augmenting Expression of Coinhibitory Molecule in LdCen^−/−^ Infected DCs

Next, we investigated the role of TLR-9 and its downstream NF-κB signaling pathway in eliciting DC function during *LdCen^−/−^* infection and compared it to *LdWT* infection. We assessed the expression of costimulatory and coinhibitory molecules in *LdCen^−/−^* infected DCs under either TLR-9 silenced or JSH-23 treated/untreated conditions and compared them to *LdWT* infection by flow cytometry. The strategy for gating and representative individual flow plots for one costimulatory (MHCII) as well as one coinhibitory (CD200) molecule have been shown in Figure 3A,B. *LdCen^−/−^* infected DCs showed a significant increase of co-stimulatory molecules such as MHCII (Figure 3A,C), CD40 (Figure 3D), and CD80 (Figure 3E) compared to *LdWT* infected mice under control siRNA or JSH-23 untreated conditions. A substantial decrease in the expression of MHCII, CD40, and CD80 was seen after TLR-9 silencing or JSH-23 treatment in DCs infected with *LdCen^−/−^* but not in *LdWT* infected DCs (Figure 3A,C–E). Additionally, we measured the expression of co-inhibitory molecules such as CD200 in both *LdWT* or *LdCen^−/−^* infected DCs under either TLR-9-silenced or JSH-23 treated conditions. *LdCen^−/−^*infected DCs exhibited significantly reduced expression of CD200 (Figure 3B,F) compared to *LdWT* under control siRNA or JSH-23 untreated conditions. On the contrary, TLR-9 silencing or NF-κB inhibition significantly increased CD200 in *LdCen^−/−^* infected DCs albeit it had no effect in *LdWT-*infected DCs (Figure 3B,F).

### 3.4. TLR-9 Silencing or Inhibition of NF-κB Transcriptional Activity Significantly Reduced LdCen^−/−^ Infected DC Mediated T Cell Proliferation and Altered Cytokine Expression

We analyzed the effect of TLR-9 silencing or inhibition of NF-κB transcriptional activity on the functional activity of DCs. We specifically examined the capacity of parasite antigens to be presented by infected DCs to CFSE-labeled naïve CD4^+^T cells to determine their antigen presentation capabilities under TLR-9 siRNA or JSH-23 treated/untreated conditions (Figure 4A). *LdCen*^−/−^ infected DCs significantly increased antigen-specific CD4^+^T cell proliferation after 5 days of co-culture compared to those co-cultured with DCs from uninfected or *LdWT* infected under control siRNA or JSH-23 untreated conditions (Figure 4A). Following TLR-9 silencing/NF-κB inhibition, antigen-specific CD4^+^ T cell proliferation co-cultured with DCs from uninfected and *LdWT-*infected cultures did not vary (Figure 4A). However, the proliferation of Ag-specific CD4^+^T cells was significantly reduced in *LdCen^−/−^* infected DCs compared to the same population from control siRNA or JSH-23 untreated *LdCen^−/−^* infected DCs (Figure 4A). The generation of cytokines in the DC-CD4^+^ T cell co-cultures’ supernatants was then evaluated and showed that *LdCen^−/−^* infected DC with CD4T cell had a considerable increase in IL-12 p40 (Figure 4B) and TNF-α (Figure 4C) along with a significant decrease in IL-10 (Figure 4D) production compared to *LdWT* infection under control siRNA or JSH-23 untreated condition. Conversely, following TLR-9 silencing or NF-κB inactivation, there was a significant decrease in IL-12p40 (Figure 4B) and TNF-α levels (Figure 4C) while IL-10 levels were significantly elevated (Figure 4D) compared to control siRNA or the JSH-23 untreated condition. In contrast, TLR-9 siRNA or JSH-23 itself had no effect on the cytokine levels from *LdWT-*infected DC-CD4T cell co-cultures (Figure 4B–D).

### 3.5. Absence of TLR-9 Abrogates LdCen^−/−^ Induced Protection against LdWT Infection

To ascertain the role of TLR-9 signaling in *LdCen^−/−^* mediated protection against virulent *Leishmania* challenge, wild type (WT) or TLR-9^−/−^ mice were immunized with *LdCen^−/−^* for 21 days (3 weeks). The schematic for the treatment regimen has been shown in Figure 5A. TLR-9 mRNA expression in immunized WT and TLR-9^−/−^ mice was determined in total spleen cells at day 22. TLR-9 mRNA expression was significantly higher in splenic cells at day 22 in immunized WT mice compared to TLR-9^−/−^ mice (Figure 5B) confirming the abrogation of the TLR-9 gene in TLR-9^−/−^ mice. Further, a significantly reduced IFN-γ:IL10 ratio in splenocytes from *LdCen^−/−^* immunized TLR-9^−/−^ mice compared to WT immunized mice at 3 weeks post-immunization was observed (Figure 5C). At day 22 post-immunization, mice were challenged with virulent *Leishmania donovani* parasites and were monitored for 6 weeks. Age-matched naïve mice were challenged with virulent *L. donovani* parasites (i. v.). Significantly reduced IFN-γ: IL10 ratio in splenocytes from *LdCen^−/−^* immunized TLR-9^−/−^ mice compared to WT immunized mice after 6 weeks post-challenge indicated a poor Th1 response (Figure 5C). Parasite burden in the spleen, as well as liver after 6 weeks of challenge with *LdWT* parasites, showed the absence of TLR-9 reversed *LdCen^−/−^* mediated parasite control as evidenced by substantially increased parasite burden at the 6-week post-challenge period in comparison to immunized challenged WT mice (Figure 5D,E). Thus, activation of the TLR-9 pathway is important for the *LdCen^−/−^* induced protective immunity.

## 4. Discussion

TLRs, mostly found on phagocytes, play a crucial role in protecting the host against various pathogens [3,4,5,6,7]. During *Leishmania* infection, TLRs such as 2, 3, 4, 7, and 9 are essential for initiating an immunological defense response [10,11,12,49]. The addition of TLR ligands has been shown to enhance immunogenicity in experimental vaccination using soluble, heat-killed, or recombinant antigens from various *Leishmania* species, such as *L. donovani*, *L. major*, and *L. amazonensis* [39,40,41,42,43,50]. The live attenuated *L. donovani* parasite vaccine (*LdCen^−/−^*) without exogenous addition of TLR ligands has demonstrated efficacy in preclinical investigations, resulting in the induction of protective immunity in experimental animal models [31,32,33,34]. Therefore, we wanted to explore if *LdCen^−/−^* immunization inherently engages various TLRs towards inducing protective immunity.

Macrophages and DCs, two important components of the innate immune system, detect pathogens via TLRs. This activation is critical in initiating both the innate and acquired immune response [38]. Our previous studies have shown that after infection with *LdCen^−/−^* parasites, macrophages and DCs acquire a pro-inflammatory phenotype [28,36], which could be due to the recognition of parasite ligands by various TLRs. In this study, we analyzed the expression of TLRs 2, 4, and 9 in macrophages and DCs in response to *LdCen^−/−^* and *LdWT* infections and found an elevated expression of TLR-9 transcript in *LdCen^−/−^* infection compared to *LdWT* infection. Notably, DCs have been shown to be important in activating TLR-9 through its ligand, CpG oligodeoxynucleotides, and serve as an adjuvant for a vaccine against *L. major* [44]. Therefore, in the present study, we determined the function of TLR-9 and the signaling molecules that it triggers exclusively in DCs during *LdCen^−/−^* infection. We observed, upon *LdCen^−/−^* infection, enhanced expression of TLR-9 receptor on DCs stimulated NF-κB signaling protein through MyD88-dependent mechanism to produce proinflammatory cytokines such as IL-12 and TNF, which have a protective function during *Leishmania* infection. Further, by examining the signaling events, we determined that silencing either MyD88 or TLR-9 reduces the production of proinflammatory cytokines in DCs during *LdCen^−/−^* infection. This suggests that the generation of pro-inflammatory responses in *LdCen^−/−^*infected DCs is enabled by the activation of the MyD88 aided by TLR-9 downstream signaling. Notably, MyD88 assisted activation of TLR-9 signaling in DCs has been reported during *L. infantum* infection [3]. Our results also emphasize the importance of TLR-9 and NF-κB in *LdCen^−/−^* infection induced DC activation. The inactivation of either TLR-9 or NF-κB prevented the activation and maturation of DCs and the subsequent presentation of antigens to CD4^+^ T cells. Additionally, the considerably higher parasite load in *LdCen^−/−^* immunized TLR-9^−/−^ animals after challenge further shows the impairment of vaccination immunity in the absence of TLR-9 signaling and highlights the importance of TLR-9 in *LdCen^−/−^* vaccine-induced immunity. Similarly, a recent study also showed that the TLR-9-binding *L. amazonensis* antigen vaccine *LaAg* comprising CpG motifs was ineffective in protecting against virulent challenges in TLR-9 deficient mice because of the lack of a protective Th1 response [51].

Studies have investigated TLR-9 expression during *L. donovani* infection, but the findings have been inconsistent. One study showed no variation in TLR-9 mRNA expression in splenic biopsies and PBMCs of VL patients’ before and after treatment, indicating that TLR-9 activation might be inhibited and not significant during *L. donovani* infection [52]. Thus, TLR-9 activation might not be essential in virulent infection but could play a crucial role in *LdCen^−/−^* vaccine-induced immunity. A separate study found a substantial increase in TLR-9 expression in whole blood samples from Sudanese VL patients. However, these results were considered inconclusive since they did not align with TLR-9′s established role in providing protection [53]. Nevertheless, it is important to note that in all these studies mentioned above, TLR-9 expression was studied during the active disease whereas our studies investigated TLR-9 expression in APCs at an early stage of infection, suggesting that there can be functional differences of TLR-9 during active disease verses in early stages of developing immune response.

Studies have shown that the detection of *L. donovani* CpG DNA by TLR-9 [54] can trigger a Th1-mediated immune response [55], and the anti-leishmanial effect of miltefosine in infected THP1 cells or peripheral blood mononuclear cells from VL patients has also been linked to proinflammatory responses driven by TLR-9, demonstrating that TLR-9 is necessary for parasite control [56]. Therefore, it is reasonable to suggest that the upregulation of TLR-9 mRNA expression and the consequent activation of downstream signaling in *LdCen^−/−^*infected DCs could be essential in the proinflammatory response triggered by the vaccination. This response might help to confer protection against a virulent challenge as observed in our study.

The impact of TLR-9 on infections caused by various species of *Leishmania* is variable and depends on the specific parasite species. For example, the activation and maturation of dendritic cells (DCs) during *L. major* infection is influenced by TLR-9 signaling. This leads to an increase in IFN-γ production by CD4^+^T cells, which in turn enhances the wound healing process during the infection. Thus, activation of DCs by TLR-9 and subsequent development of Th1 cells is a crucial factor in the overall immune response to *L. major* infection. [19]. Likewise, infection with *L. infantum* activates a protective type-I interferon response and production of IL-12 through TLR-9 activation [21]. However, *L. guanenesis*, which has an endogenous RNA virus, fails to activate TLR-9-Myd88, leading to increased IL-4 and IL-13 and reduced IL-12 levels, making the host more susceptible to disease [57]. Thus, the role of TLR-9 in the control of parasite infections is not unequivocal and may result in either protective or susceptibility responses depending on the *Leishmania* species.

Besides increased expression of TLR-9 upon *LdCen^−/−^* immunization, we observed decreased expression of TLR-2 in APCs compared to *LdWT* infection. This suggests that *LdWT* parasites may use LPG-TLR-2-dependent Th2 bias to weaken the host effector response, allowing the parasite to grow within the host, as has been shown during *L. major* or *L. donovani* infection [58,59]. Further, TLR-2 activation during *L. major* infection decreases the expression of TLR-9 and reduces anti-leishmanial responses [58]. Thus, reduced expression of TLR-2 in macrophages and DCs from mice infected with *LdCen^−/−^* likely led to increased expression of TLR-9. Thus, *LdWT* and *LdCen^−/−^* parasites manipulate DC function through different TLR signaling pathways. Even though TLR-4 is essential for causing antileishmanial activity in macrophages during experimental *L. donovani* infection [59], in our work, we did not detect any discernible differences in TLR-4 expression between *LdWT* and *LdCen^−/−^* infection. It appears likely that TLR-4-mediated activation is not necessary for vaccine-induced immunity.

Collectively, these in vitro and in vivo studies point towards a novel mechanism of TLR-9 mediated immunoprotection during *LdCen^−/−^* infection. Recent studies have hypothesized that activating TLR-9 with parasitic CpG islands might serve as possible adjuvants, especially against VL [54,60]. Further, the conventional *Leishmania* vaccines such as *Leishmania* soluble antigen, recombinant vaccines, heat-killed vaccines, and DNA vaccines require the presence of immunostimulatory TLR-9 ligand CpG-ODN to render protection against leishmaniasis [39,40,41,42,43,44,50]. In contrast, live attenuated *LdCen^−/−^* vaccine inherently elevates TLR-9 expression, eliminating the need for adjuvants and making it a more affordable option for vaccine formulation against different types of leishmaniasis. These findings have broad implications for the future development of vaccines against leishmaniasis and highlight the importance of the TLR-9 pathway in the innate immune response against this disease.

## Figures and Tables

**Figure 1 pathogens-12-00534-f001:**
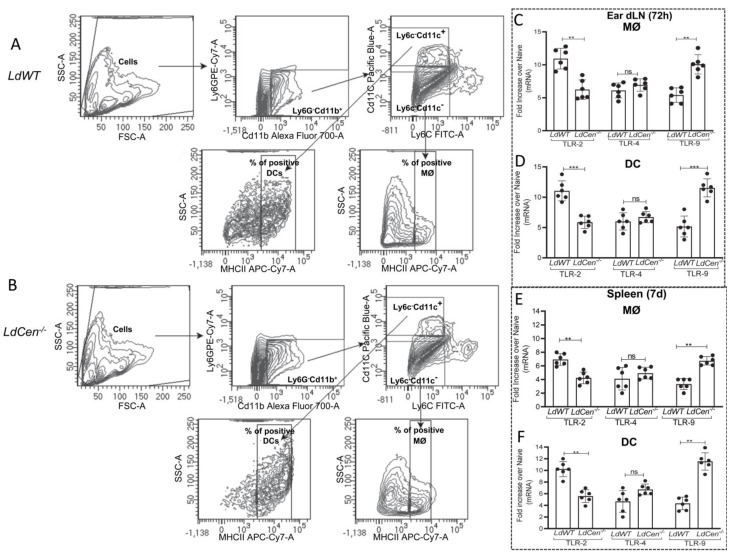
*LdCen*^−/−^ infection induces TLR-9 expression in ear dLN and spleen-derived macrophages and dendritic cells. Macrophages (MØ) (Cd11b^+^Ly6C^−^Ly6G^−^CD11c^−^MHCII^+^) and DCs (Cd11b^+^Ly6C^−^Ly6G^−^CD11c^+^MHCII^hi^) were flow sorted from the ear dLN or spleen 72 h and 7 d post infection, respectively. (**A**,**B**) Sorting strategy showing the percentage of the positive population of macrophages and DCs from ear dLN of *LdWT* and *LdCen^−/−^* infected mice. (**C**–**F**) mRNA expression levels of TLR-2, 4, and 9 from sorted macrophages and DCs in (**C**,**D**) ear dLN and (**E**,**F**) spleen were estimated by qPCR as described in Materials and Methods and expressed as fold change from uninfected naive mice. The experiment was repeated three times with pooled digests from five to six ear dLNs per experiment. The data represent the mean values ± SD of results from three independent experiments that all yielded similar results (n = 6). ** *p* < 0.005; *** *p* < 0.0005 between the groups.

**Figure 2 pathogens-12-00534-f002:**
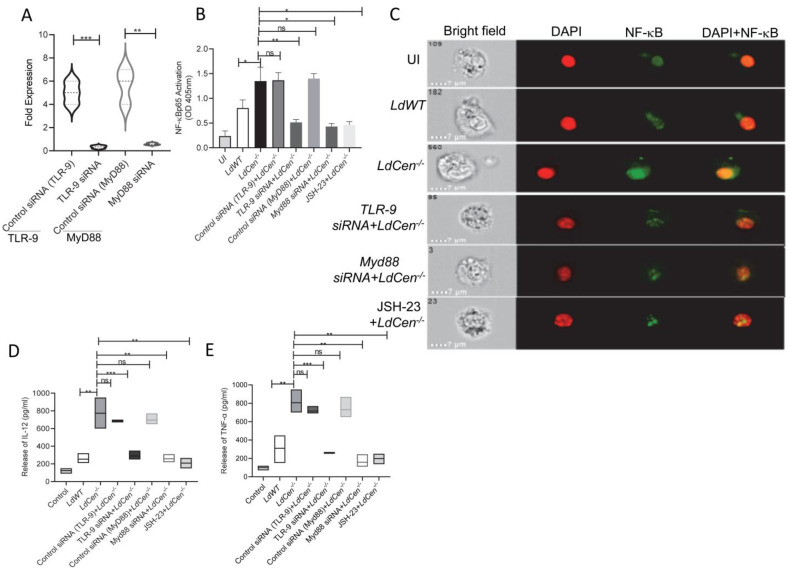
*LdCen^−/−^* infected BMDCs showed heightened NF-kB activation and proinflammatory cytokine generation via the TLR-9-Myd88 pathway. (**A**) BMDCs were transfected with control siRNA (TLR-9)/TLR-9 siRNA or control siRNA (MyD88)/MyD88 siRNA or treated with JSH-23 followed by treatment with LPS (1 μg/mL) for 45 min and infected with *LdCen^−/−^* parasites for 6 h. The transfection efficiency of TLR-9 siRNA/Myd88 siRNA was checked for every sample by qPCR as described in Materials and Methods and the representative data for *LdCen^−/−^* infected group has been shown. The data represent the mean values ± SD of results from three independent experiments that all yielded similar results. ** *p* < 0.005; *** *p* < 0.0005 between the groups. (**B**) Cell extract was prepared, and NF-κB p65 activation was measured quantitatively using Trans AM NF-κB kit. Data for NF-κB p65 activation (Absorbance at 450 nM) are expressed as means ± standard deviations (SD) from triplicate experiments that yielded similar results. * *p* < 0.05; ** *p* < 0.005. (**C**) LPS (1 μg/mL) treated BMDCs were either infected with *LdWT* parasites or transfected with TLR-9 siRNA or MyD88 siRNA or treated with JSH-23 and infected with *LdCen^−/−^* parasites for 6 h, and nuclear translocation was analyzed in Flowsight. Representative images of treated cells. For improved visualization, red (DAPI) and green (NF-κB) colors were assigned in IDEAS software. Brightfield (BF), co-localization (yellow). (**D**,**E**) LPS (1 μg/mL) treated BMDCs were either infected with *LdWT* parasites or transfected with control siRNA (TLR-9)/ /TLR-9 siRNA or control siRNA (MyD88)/ /MyD88 siRNA or JSH-23 and infected *LdCen^−/−^* parasites for 24 h. Culture supernatants were collected to determine the (**D**) IL-12 and (**E**) TNF- α release by ELISA. The data represent the mean values ± SD of results from three independent experiments that all yielded similar results. * *p* < 0.05; ** *p* < 0.005; *** *p* < 0.0005 between the groups.

**Figure 3 pathogens-12-00534-f003:**
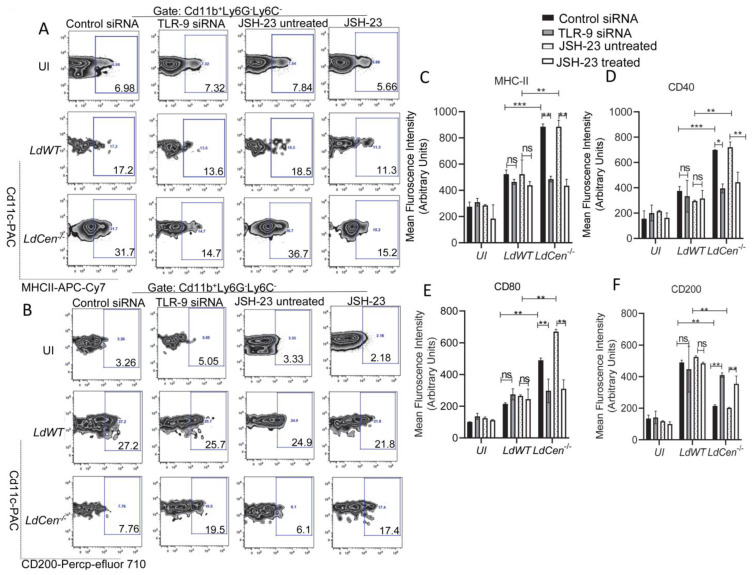
TLR-9 silencing or inhibition of NF-kB transcriptional activity significantly reduced the expression of activation marker and costimulatory molecule expression while augmenting coinhibitory molecule expression in *LdCen^−/−^* infected DCs. BMDCs were transfected with control siRNA/TLR-9 siRNA or treated/untreated with JSH-23 followed by infection with *LdWT/LdCen^−/−^* parasites for 24 h. The expression of MHCII, CD40, CD80, and CD200 in the BMDCs was analyzed by flow cytometry. (**A**,**B**) The gating strategy and the individual flow plots for (**A**) MHCII and (**B**) CD200 have been shown. (**C**–**F**) Mean fluorescence intensity of (**C**) MHCII, (**D**) CD40, (**E**) CD80, and (**F**) CD200 expression in BMDCs have been represented by the bar diagram. The data represent the mean values ± SD of results from three independent experiments that all yielded similar results (n = 6). * *p* < 0.05; ** *p* < 0.005; *** *p* < 0.0005 between the groups.

**Figure 4 pathogens-12-00534-f004:**
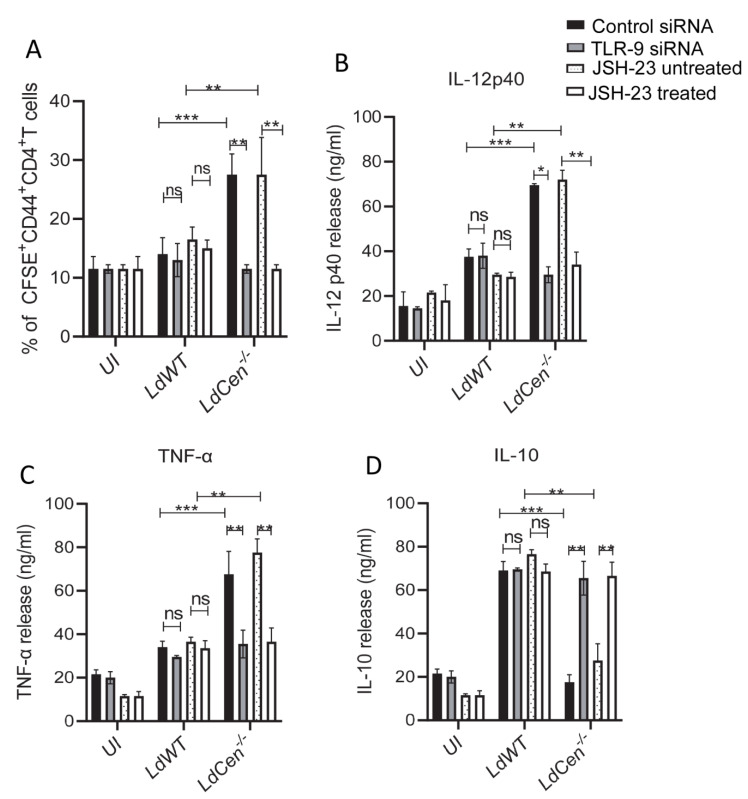
TLR-9 silencing or inhibition of NF-kB transcriptional activity significantly reduced *LdCen^−/−^* infected DC mediated T cell proliferation and generates Th2 response. (**A**) BMDCs were transfected with control siRNA/TLR-9 siRNA or treated//untreated with JSH-23, pulsed with OVA peptide followed by infection with *LdWT/LdCen^−/−^* parasites for 24 h, and then co-cultured with purified CFSE-labeled CD4+T cells from DO11.10 transgenic mice for 5 d. OT-II T cells proliferation after 5 d culture with OVA peptide-pulsed DCs was estimated by flow cytometry by studying CFSE dilution of gated CD4^+^ T cells and is represented by the bar diagram. Cell proliferation was analyzed in triplicate experiments. The data represent the mean values ± SD of results from three independent experiments that all yielded similar results. ** *p* < 0.005; *** *p* < 0.0005 between the groups. (**B**–**D**) The cytokines released from the co-culture experiment were measured by sandwich ELISA as described in Materials and Methods. The data represent the mean values ± SD of results from three independent experiments that all yielded similar results. * *p* < 0.05; ** *p* < 0.005; *** *p* < 0.0005 between the groups.

**Figure 5 pathogens-12-00534-f005:**
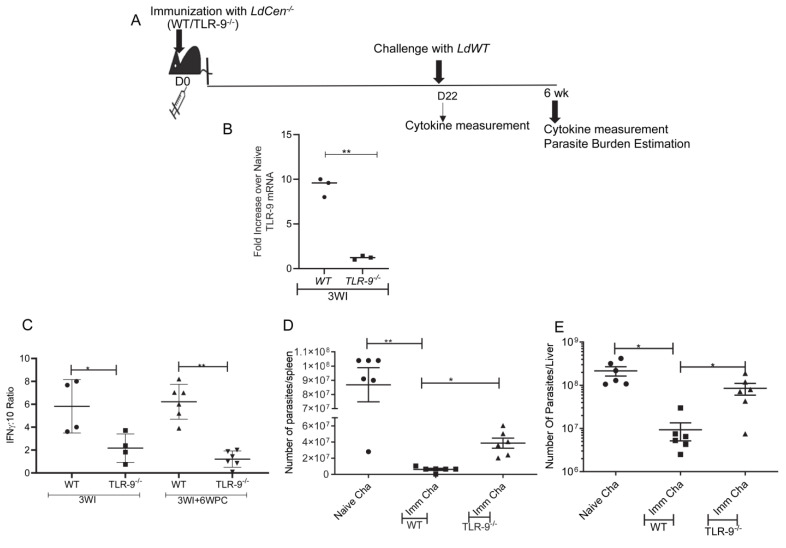
Absence of TLR-9 abrogates the *LdCen^−/−^* induced host protective immunity. (**A**) Schematic diagram showing the treatment regimen. (**B**) TLR-9 mRNA expression levels in the spleens of immunized WT and TLR-9^−/−^ mice at day 21 (3WI) was determined by qPCR as described in Materials and Methods and expressed as fold change from uninfected naive mice. The data represent the mean values ± SD of results from three independent experiments that all yielded similar results (n = 3). ** *p* < 0.005 between the groups. (**C**) Leishmania Ag–specific cytokines were measured from splenocytes of WT or TLR-9^−/−^ *LdCen*^−/−^ immunized mice at the time of challenge (3 WI, 3 wk post-immunization) and after challenge (3 WI+ 6 WPC: 6 wk post-challenge) by sandwich ELISA. The ratio of IFNγ/IL-10 is shown. The data represent the mean values ± SEM of results from two independent experiments. The mean and SEM of 4–6 mice in each group are shown. ** *p* < 0.005. (**D**) Splenic and (**E**) liver parasite burden were measured at 6 wk post-challenge in different groups of immunized-challenged and naive-challenged mice. The data represent the mean values ± SEM of results. The mean and SEM of 6 mice in each group are shown. * *p* < 0.05; ** *p* < 0.005 between the groups.

## Data Availability

All relevant data are within the manuscript.

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
