# Peer review of "Toll-like Receptor-9 (TLR-9) Signaling Is Crucial for Inducing Protective Immunity following Immunization with Genetically Modified Live Attenuated Leishmania Parasites"

_pathogens, 2023, doi:10.3390/pathogens12040534_

Round 1
Reviewer 1 Report
Manuscript “pathogens-2265183” emphasized on role of TLR-9 signaling to aggravate the protective immunity against virulent L. donovani infection.
Although, paper is well written and presented the figures and results impactful manner but there are some minor changes are needed to incorporate here like-
1. Don’t repeat those key words which are already used in article title, for example - Toll like receptor- 9, Live attenuated Leishmania.
2. Some of the abbreviations have not been extended like BMDC, Knockout symbol (-/-) and other notions keep the same appearance everywhere (text and figures) and superscripted.
3. Figure 1A and 1B if not represented/mentioned the percentage of positive population, then it seems like repetition of dot plots.
4. Keep the same y-axis for figure 1E and 1F and retain the same fonts in all figures throughout.
5. Above are some examples, I hope authors will take care of rest other similar types of mistakes at their end.
Overall, I appreciate the work done by authors, only some proofreading and minor corrections are required to article.
All the Best,
Reviewer 2 Report
In this study, the authors explored the function of TLR9 in response to LdCen -/- infection. Using RT-qPCR, FACS, ELISA, confocal and mouse models, they provide evidence that TLR9 mediated immunoprotection during LdCen -/- infection. Although the findings are interesting, more in-depth analyses are needed.
1. It is well known that TLR9 signals through different cellular compartments induce either MyD88-IRF7dependent type I IFN or MyD88-NF-κB-dependent inflammatory cytokines. Whether the production of IFN-α/ β is affected in BMDCs after infection with LdCen -/- ? The authors should detect and compare IFN-α/ β expression in BMDCs infected with LdCen -/- or LdWT.
2. Given eukaryotic DNA contains unmethylated CpG-DNA motifs, which might cause TLR9-dependent stimulation of immune cells. Fakher et al. previously reported that the recognition of L. major DNA by TLR9 promotes dendritic cell (DC) activation, which induces a Th1-dominant response that resolves the lesions. What components of LdCen -/- trigger TLR9 signaling activation?
3. Related to Figure 2C, the signal intensity looks similar in UI and LdWT groups, whereas LdCen -/- significantly promoted nuclear translocation of NF-κB. What makes LdCen -/- induce a strong signal? The authors mentioned that LdCen-/- lacks the Centrin1 gene. What’s the function of the Centrin1 gene?
4. Related to Figure 5, since Leishmania targets multiple organs, such as the spleen, liver, and lymph nodes, in addition to detecting parasite burden in the spleen, it would be better to measure the parasite burden in the liver.
